# Serum sodium on admission affects postoperative in-hospital mortality in acute aortic dissection patients

Pengfei Huang[1☯], Hongyan Wang[1☯], Dong Ma[1,2]*, Yongbo Zhao[3]*, Xiao Liu[3], Peng Su[1], Jinjin Zhang[1], Shuo Ma[1], Zhe Pan[1], Juexin Shi[1], Fangfang Hou[1], Nana Zhang[1], Xiaohui Zheng[1], Nan Liu[1], Ling Zhang[1]

1 School of Public Health, North China University of Science and Technology, Tangshan, Hebei, China,
2 Department of Biochemistry and Molecular Biology, Key Laboratory of Neural and Vascular Biology, Ministry of Education, Hebei Medical University, Shijiazhuang, Hebei, China, 3 Cardiac Surgery Department, The Fourth Hospital of Hebei Medical University, Shijiazhuang, Hebei, China

☯ These authors contributed equally to this work.
* mamamadong@163.com (DM); zhaoyongboyueyue@163.com (YZ)

**Data Availability Statement:** All relevant data are within the paper and its Supporting Information files.

## Abstract

### Background

Acute aortic dissection (AAD) is very fatal without surgical treatment. Higher serum sodium can increase in-hospital mortality of many diseases; however, the effect of serum sodium on postoperative in-hospital mortality in AAD patients remains unknown.

### Methods

We collected a total of 415 AAD patients from January 2015 to December 2019. Patients were classified into four categories (Q1-Q4) according to the admission serum sodium quartile. The cox proportional hazards model evaluated the association between serum sodium and in-hospital mortality. All-cause in-hospital mortality was set as the endpoint.

### Results

By adjusting many covariates, cox proportional hazards model revealed the in-hospital mortality risk of both Q3 and Q4 groups was 3.086 (1.242–7.671, $P = 0.015$) and 3.370 (1.384–8.204, $P = 0.007$) respectively, whereas the risk of Q2 group was not significantly increased. Univariate and multiple Cox analysis revealed that Stanford type A, serum glucose, α-hydroxybutyrate dehydrogenase and serum sodium were risk factors correlated with in-hospital death in AAD patients.

### Conclusion

The study indicates that the admission serum sodium of AAD patients has a vital impact on postoperative hospital mortality.

**Funding:** Our gratitude goes to the National Natural Science Foundation of China (No.81700416) supported for this study. The funders had no role in study design, data collection and analysis, decision to publish, or preparation of the manuscript.

**Competing interests:** The authors declare no competing interests.

## Introduction

Acute aortic dissection (AAD) has a high mortality rate, and the mortality risk for AAD patients increases by 1% per hour before medical and surgical intervention [1]. Presently, the best way to reduce mortality rate for AAD patients is timely operation. Despite increasingly rapid diagnosis and surgical management, the in-hospital mortality rate for AAD remains at >30% [2]. In addition, due to the change of economic level and residents' eating habits recently, the incidence rate of AAD remains rising.

Hypernatremia, defined as >145 mmol/L sodium levels, was reported in 2% of patients admitted to the emergency department [3]. The abnormal increase of serum sodium levels is positively associated with the mortality risk of patients with various diseases, like severe craniocerebral injury [4], aneurysm subarachnoid hemorrhage [5], and nervous system disease [6]. A study of an unrestricted hospitalized adult population in the United States reported that >142 mmol/L serum sodium concentration has been associated with mortality and different degrees of hypernatremia including mild, middle, and higher levels are associated with in-hospital mortality [7]. Even in a healthy population, higher levels of plasma sodium starting from 138 mmol/L were associated with a higher risk of mortality, especially when the sodium level exceeded 145 mmol/L [8, 9]. However, no study on the relationship between serum sodium upon admission and in-hospital mortality of AAD patients exists, since electrolyte disturbances in emergency AAD patients can easily be ignored. Therefore, this study aims to discuss whether the serum sodium associates with the postoperative in-hospital mortality of AAD patients.

## Methods

### Study population

This study was designed as a retrospective observational study that continuously collected data on AAD patients who underwent surgery in the Fourth Hospital of Hebei Medical University from January 2015 to December 2019. All patients were diagnosed by computed tomographic angiography (CTA) or magnetic resonance angiography (MRA), and medical history. The aortic dissection is considered as AAD if the time from the onset of the symptom to operation is within 14 days. Exclusion criteria included connective tissue diseases, pregnancy, traumatic dissection, and infection diseases as well as the severe lack of clinical data which is defined as the absence of complete hospitalization records, involving lack of blood glucose, blood pressure and other important indicators associated with the known risk factors of AAD. Except for basic information and indicators associated with the known risk factors of AAD, some other defect data were considered to be a moderate missing for some individual patients, who were included in the study.

All clinical data related to the study were obtained after approved by the ethics committee of the Fourth Hospital of Hebei Medical University. As patient data were anonymized, the ethics committee of the Fourth Hospital of Hebei Medical University waived the written informed consent. All procedures followed were in accordance with the revised Declaration of Helsinki.

### Clinical data collection

Clinical variables of enrolled patients with AAD was obtained through review of medical records, including gender, age, Stanford type, medical history (hypertension, diabetes, coronary artery disease, prior surgery, prior trauma, smoking and drinking), vital signs on admission (systolic blood pressure, SBP; diastolic blood pressure, DBP; heart rate) and laboratory data on admission (alanine transaminase, ALT; aspartate transaminase, AST; creatinine; urea nitrogen;

serum glucose; α-hydroxybutyric dehydrogenase, α-HBDH; white blood cell count, WBC; neutrophil count, serum sodium, et al.) as well as the length of in-hospital [10]. Smoking is defined as at least 1 cigarette daily for 1 year or more, and alcohol consumption is defined as at least once per week for a period of one month or longer. Notably, the determination of serum sodium and other ions was done by automatic electrolyte analyzer (Shenzhen Kangli AFT— 500D), with high accuracy and good precision (the CV value of Na detection was generally 0.60%), which is sufficient to meet the accuracy of this research. Stanford type A aortic dissections involve the ascending aorta whereas Stanford type B dissections involve the descending but not the ascending aorta [11]. The laboratory results were obtained using the patients' first venous blood samples taken on admission to the hospital. The principles and strategies of surgical techniques were determined by experienced surgeons. The outcomes of in-hospital patients with AAD were gathered from medical records. All-cause mortality during hospitalization was defined as the endpoint.

## Statistical analysis

According to the quartile of serum sodium concentration on admission, the patients were divided into four groups (quartile 1- quartile 4, Q1- Q4). Continuous variables were expressed as mean ± SD or median (quartile range). Categorical variables were expressed as the number of patients (%). One-way analysis of variance (ANOVA) test and Kruskal-Wallis H test were used to compare the difference of continuous variables between quartile groups. The chi-squared test or Fisher's exact test was used to compare categorical variables. Proportional hazards assumption (PH) and Schoenfeld residual method were performed to test the availability of Cox proportional risk model. Univariate and multivariate cox proportional hazards model was employed to evaluate the association between serum sodium and in-hospital mortality. We constructed three models: crude model, with no adjustment of covariates; model 1, adjusted for age and gender; and model 2 including other covariates presented in Tables 3 and 4 [12]. Survival curves were constructed using the Kaplan–Meier method estimates and compared with the log-rank test. All the analyses were carried out with SPSS 25.0, two sided $P$ values < 0.05 were considered statistically significant.

## Result

### Baseline characteristics of selected participants

A total of 415 AAD patients were enrolled in the present study based on the inclusion and exclusion criteria (Fig 1). Table 1 shows patient characteristics according to the survival situation at the time of discharge. Among the 415 patients, 61 (14.70%) died after surgery during hospitalization, and 354 (85.30%) survived. The SBP (130.07 ± 23.87 vs. 140.5 ± 28.91 mmHg, $P$ = 0.008) and the DBP (75.28 ± 18.00 vs. 81.25 ± 19.39 mmHg, $P$ = 0.025) of the non-survivors were lower than the survivors. The percentage of Stanford type A in the non-survivor group was more than that in the survivor group (93.44 vs. 56.78%, $P$ < 0.001). The length of the in-hospital stay was significantly shorter in the death group than in the survival group (2 [1, 9] vs. 14 [7, 21] days, $P$ < 0.001). AST (43.20 [24.05, 81.75] vs. 29.75 [19.00, 49.80] U/L, $P$ = 0.006), creatinine (105.00 [68.00, 141.50] vs. 77.50 [62.00, 104.25] μmol/L, $P$ = 0.004), and serum glucose (7.76 [6.43, 10.37] vs. 7.13 [5.98, 8.53] mmol/L, $P$ = 0.009) levels were significantly higher in the death than in the survival group. α-HBDH (206.00 [165.80, 298.70] vs. 258.30 [173.00, 371.40] U/L, $P$ = 0.029) was lower in the death than in the survival group. Serum sodium (140 ± 6.55 vs. 138 ± 4.38 mmol/L, $P$ = 0.006) and serum chloride (105.70 [103.00, 110.00] vs. 104.00 [101.00, 107.00] mmol/L, $P$ = 0.002) levels were also higher in the death than in the survival group.

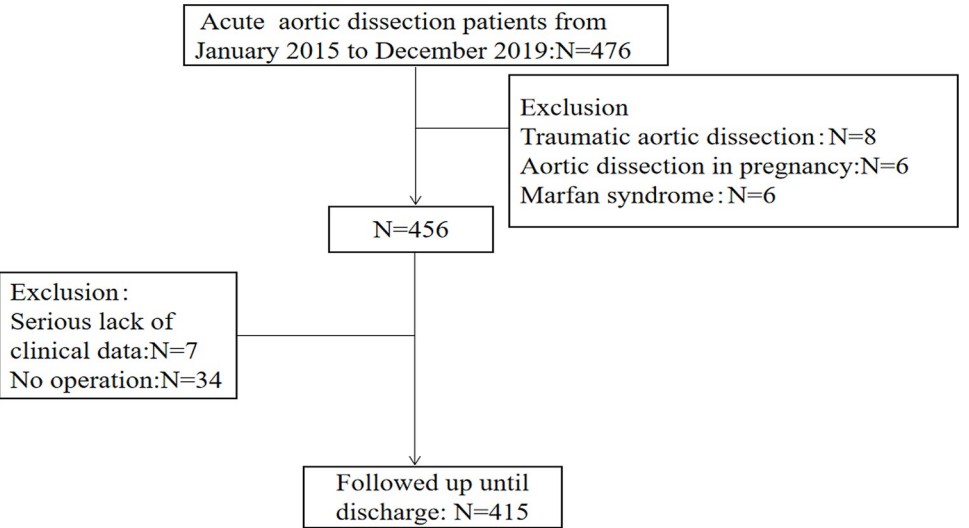

**Fig 1. Flow chart of patient enrollment.**

Moreover, across all quartiles (Q1: ≤ 136 mmol/L, Q2: 137–138 mmol/L, Q3: 139–140 mmol/L, Q4: ≥141 mmol/L), there were statistical differences in years, monocyte count, and serum chloride. Meanwhile, the highest serum sodium quartile group had a higher neutrophil count (Table 2).

## Survival curve analysis

To further investigate the relationship between patient survival and serum sodium in AAD, a comparison of Kaplan–Meier curves with the different serum sodium levels was conducted. Fig 2 shows Kaplan–Meier survival curves that proved a decreased short-term survival associated with admission serum sodium in AAD patients (from Q1 to Q4 groups, log-rank test $P = 0.010$).

## In-hospital mortality analysis

Next, the in-hospital mortality of AAD patients was further analyzed in accordance with the admission serum sodium quartiles. Fig 3 shows that the significant increase of in-hospital mortality was accompanied by a gradual elevation in serum sodium levels in admitted AAD patients (Q1, 7.02%; Q2, 11.65%; Q3, 18.62%; Q4, 22.92%; $P = 0.006$).

Basing on the Q1 group as a reference, in-hospital mortality risk slightly increased in the Q2 group, however, it was significantly increased by 2.890-fold with Q3 and by 3.253-fold with Q4, respectively.

Following the adjustment for gender and years (model 1), the death risk of the Q3 and Q4 groups was still 2.825-fold (95% CI, 1.228–6.502-fold; $P = 0.015$) and 3.241-fold (95% CI, 1.441–7.290-fold; $P = 0.004$), respectively, while a 1.728-fold elevation in the Q2 group also occurred (95% CI, 0.705–4.233; $P = 0.232$).

More importantly, following the adjustment for gender, age, Stanford type, hypertension, diabetes, coronary artery disease, smoking, drinking, serum glucose, and α-HBDH (model 2), the in-hospital mortality risk was further elevated in Q3 (HR = 3.086, 95% CI, 1.242–7.671; $P = 0.015$) and Q4 (HR = 3.370, 95% CI, 1.384–8.204; $P = 0.007$), respectively, although the

**Table 1. Baseline characteristics of patients with AAD.**

| Variable | Non-survivor (n = 61) | Survivor (n = 354) | P |
|---|---|---|---|
| Years | 54.21±12.68 | 53.01±12.15 | 0.479 |
| Male(N,%) | 39,63.93 | 266,71.14 | 0.067 |
| Hypertension(N,%) | 46,75.41 | 258,72.88 | 0.680 |
| Diabetes(N,%) | 2,3.28 | 6,1.69 | 0.405 |
| CAD(N,%) | 4,6.56 | 14,3.95 | 0.357 |
| Smoking(N,%) | 27,44.26 | 190,53.67 | 0.174 |
| Drinking(N,%) | 31,50.82 | 202,57.06 | 0.364 |
| Stanford type A(N,%) | 57,93.44 | 201,56.78 | <0.001 |
| History of trauma(N,%) | 4,6.56 | 17,4.80 | 0.564 |
| Surgical history(N,%) | 18,29.50 | 90,25.42 | 0.502 |
| Heart rate(b.P.m) | 79±16.97 | 80.57±17.60 | 0.577 |
| SBP(mmHg) | 130.07±23.87 | 140.50±28.91 | 0.008 |
| DBP(mmHg) | 75.28±18.00 | 81.25±19.39 | 0.025 |
| length of hospital(day) | 2(1,9) | 14(7,21) | <0.001 |
| ALT(U/L) | 26.70(16.4,43.55) | 21.75(15.50,31.88) | 0.084 |
| AST(U/L) | 43.20(24.05,81.7) | 29.75(19.00,49.80) | 0.006 |
| Creatinine(μmol/L) | 105.00(68.00,141.50) | 77.50(62.00,104.25) | 0.004 |
| Urea nitrogen(mmol/L) | 7.00(5.45,9.25) | 6.10(4.70,8.10) | 0.060 |
| Serum glucose(mmol/L) | 7.76(6.43,10.37) | 7.13(5.98,8.53) | 0.009 |
| AHB(U/L) | 206.00(165.80,298.70) | 258.30(173.00,371.40) | 0.029 |
| WBC count(10*9/L) | 12.45±4.83 | 11.74±3.88 | 0.282 |
| Neutrophil count(10*9/L) | 9.56(7.47,12.57) | 9.83(7.46,12.46) | 0.570 |
| Lymphocyte count(10*9/L) | 0.87(0.67,1.08) | 0.94(0.65,1.33) | 0.291 |
| Monocyte count(10*9/L) | 0.66(0.50,0.93) | 0.67(0.51,0.92) | 0.714 |
| Serum sodium(mmol/L) | 140±6.55 | 138±4.38 | 0.006 |
| Serum potassium(mmol/L) | 4.00(3.35,4.10) | 4.00(3.40,4.00) | 0.418 |
| Serum chloride(mmol/L) | 105.70(103.00,110.00) | 104.00(101.00,107.00) | 0.002 |
| Serum calcium(mmol/L) | 2.14(2.06,2.26) | 2.18(2.10,2.28) | 0.129 |

CAD: coronary artery disease; SBP: systolic blood pressure; DBP, diastolic blood pressure; ALT: Alanine transaminase; AST: Aspartate transaminase; α-HBDH: alpha-hydroxybutyric dehydrogenase; WBC: white blood cell.

For numerical raw data, please see S1 Data.

risk in Q2 (HR = 1.718, 95% CI, 0.658–4.486; *P* = 0.269) still is not significantly increased (Table 3).

## Univariate and multiple Cox analysis for in-hospital mortality

In order to verify whether Cox proportional risk model would be available for the short-term survival analysis in AAD patients, proportional hazards assumption (PH) and schoenfeld residual method were performed to test the assumption. We plot the smooth curve residual diagram of Schoenfeld's residual with time, and the results showed no linear correlation between Schoenfeld's residual and time rank (Fig 4). Moreover, we tested that the correlation between Schoenfeld's residual and time rank, and found that Schoenfeld's residual was independent on time variables (Pearson correlation coefficient, r = -0.22, P = 0.864). Based on these two results of the smooth curve trend and the Pearson correlation from Schoenfeld's residual analysis, indicating that Cox model is suitable for the short-term survival analysis.

**Table 2. Clinical characteristics of patients of serum sodium quartiles.**

| Variable | Q1(n = 114) | Q2(n = 103) | Q3(n = 102) | Q4(n = 96) |
|---|---|---|---|---|
| Age(years)* | 51.18±12.61 | 52.50±13.11 | 56.11±11.93 | 53.21±10.53 |
| Male(N,%) | 83,72.80 | 79,76.70 | 75,73.53 | 68,79.83 |
| Hypertension(N,%) | 80,70.18 | 74,71.84 | 78,76.47 | 72,75.00 |
| Diabetes(N,%) | 1,0.87 | 3,2.91 | 1,0.98 | 3,3.13 |
| CAD(N,%) | 4,3.51 | 7,6.80 | 5,4.90 | 2,2.08 |
| Smoking(N,%) | 60,52.63 | 47,45.63 | 61,59.80 | 49,51.04 |
| Drinking(N,%) | 62,53.39 | 57,55.33 | 59,57.84 | 55,57.29 |
| Stanford type A | 63,55.26 | 67,65.05 | 59,57.84 | 69,71.88 |
| non-survivor ** | 6,9.52 | 11,16.42 | 18,30.51 | 22,31.88 |
| survivor | 57,90.48 | 56,83.58 | 42,71.19 | 47,68.12 |
| History of trauma(N,%) | 4,3.51 | 4,3.88 | 9,8.82 | 4,4.17 |
| Surgical history(N,%) | 27,23.68 | 28,27.18 | 28,27.45 | 25,26.04 |
| Heart rate(b.p.m) | 82.18±15.46 | 80.03±21.02 | 79.90±14.94 | 79.07±18.23 |
| SBP(mmHg) | 139.02±26.92 | 140.65±27.24 | 138.75±31.60 | 137.33±28.26 |
| DBP(mmHg) | 80.25±17.48 | 81.94±19.77 | 80.68±21.22 | 78.49±18.63 |
| length of in-hospital(day) | 13.46±8.62 | 12.83±8.48 | 11.54±8.94 | 13.89±9.73 |
| ALT(U/L) | 24.25(15.45,32.53) | 19.80(14.00,29.20) | 22.60(15.90,31.96) | 24.75(16.65,36.00) |
| AST(U/L) | 33.80(19.93,52.80) | 28.00(18.30,46.70) | 29.25(19.45,49.53) | 32.65(20.61,67.55) |
| Creatinine(μmol/L) | 79.50(61.75,110.00) | 80.00(63.00,111.00) | 72.50(60.75,103.50) | 84.50(65.00,122.25) |
| Urea nitrogen(mmol/L) | 6.00(4.30,7.35) | 6.80(5.00,8.90) | 5.95(4.80,7.70) | 6.60(5.40,8.70) |
| Serum glucose(mmol/L) | 7.17(6.13,9.09) | 7.11(5.97,9.32) | 7.70(5.97,8.11) | 7.40(6.32,9.15) |
| α-HBDH(U/L) | 214.90(167.50,302.95) | 208.80(160.90,311.90) | 197.90(161.95,271.60) | 233.85(182.90,328.85) |
| WBC count(10*9/L) | 11.81±4.19 | 11.51±3.61 | 11.47±3.29 | 12.64±4.86 |
| Neutrophil count(10*9/L)* | 9.93±4.04 | 9.78±3.57 | 9.71±3.28 | 11.35±4.75 |
| Lymphocyte count(10*9/L) | 0.94(0.65,1.28) | 0.96(0.67,1.40) | 0.62(0.44,0.86) | 0.94(0.65,1.23) |
| Monocyte count(10*9/L)* | 0.70(0.55,0.97) | 0.62(0.44,0.86) | 0.64(0.51,0.93) | 0.75(0.54,1.03) |
| Serum potassium(mmol/L) | 3.90(3.40,4.00) | 4.00(3.40,4.00) | 4.00(3.40,4.10) | 4.00(3.40,4.18) |
| Serum chloride(mmol/L)* | 101.00(97.99,104.00) | 105.00(102.00,106.00) | 105.85(103.00,109.00) | 107.35(104.00,110.00) |
| Serum calcium(mmol/L) | 2.16(2.09,2.24) | 2.17(2.10,2.29) | 2.20(2.10,2.28) | 2.18(2.11,2.29) |

*: $P < 0.05$

**: Stanford type A as subgroup, compared in-hospital mortality in Q1-Q4 groups, $P < 0.05$. CAD: coronary artery disease; SBP: systolic blood pressure; DBP, diastolic blood pressure; ALT: Alanine transaminase; AST: Aspartate transaminase; α-HBDH: alpha-hydroxybutyric dehydrogenase; WBC: white blood cell.

For numerical raw data, please see S1 Data.

Univariate analysis indicated that eight variables, including Stanford type A, SBP, DBP, creatinine, serum glucose, α-HBDH, serum sodium, and serum calcium, were associated with in-hospital mortality with a *P*-value of <0.10 and were put into cox regression (forward LR method) analysis, the results showing that Stanford type A (HR = 3.634, 95% CI, 1.638–8.086; *P* = 0.002), serum glucose (HR = 1.077, 95% CI, 1.025–1.132; *P* = 0.003), α-HBDH (HR = 1.001, 95% CI, 1.001–1.002; *P* < 0.001), and serum sodium (HR = 1.068, 95% CI, 1.029–1.109, *P* = 0.001) were risk factors correlated with in-hospital mortality in AAD patients (Table 4).

## Discussion

AAD is a fatal cardiovascular disease [13]. Since the underlying mechanisms of the development and progression of spontaneous AAD cases remain unclear, a screening or prevention

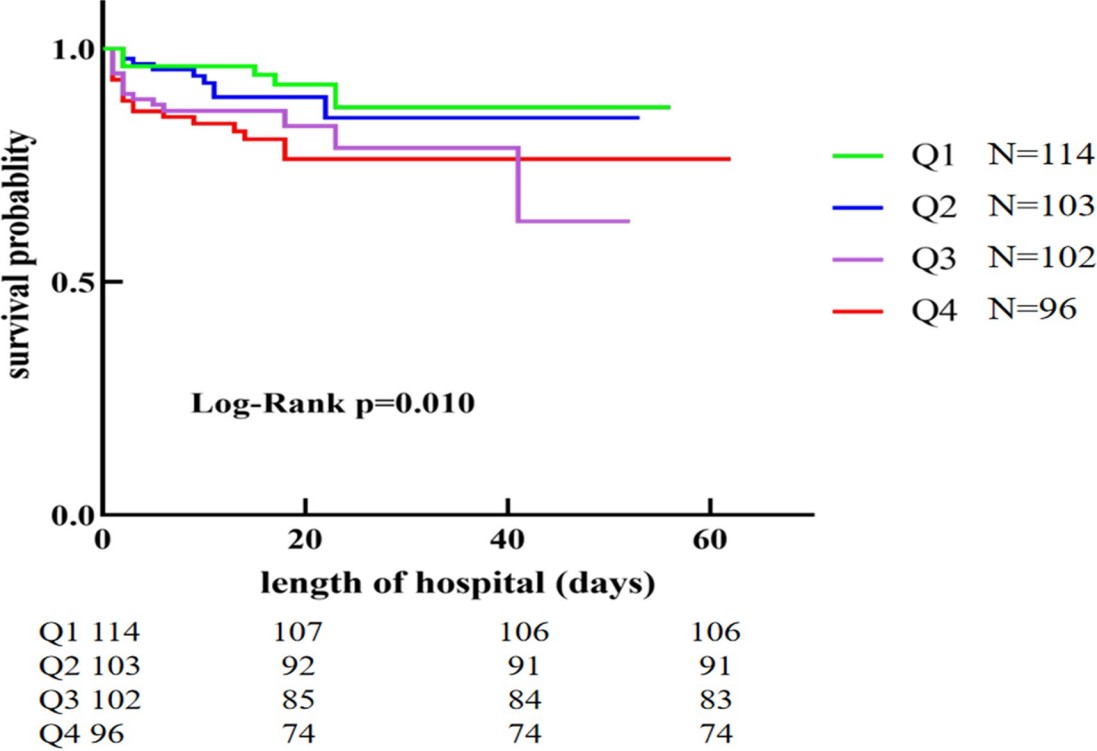

**Fig 2. Kaplan-Meier curves for the chronological trend in mortality on serum sodium quartiles groups.**

strategy for these patients has yet to be developed [14]. In previous studies, preoperative creatinine, uric acid, serum tenascin-C, and inflammatory factors were all associated with postoperative mortality following AAD [10, 15–17]. Additionally, poor organ perfusion and hemodynamic conditions, such as hypotension, shock, pericardial tamponade, pulse deficiency, and renal failure, further contribute to higher mortality in AAD patients [12]. The present study was the first to evaluate the relationship between serum sodium upon admission and in-hospital mortality of AAD patients, the results showing the increased in-hospital mortality risk in AAD patients with high-sodium levels upon admission. The higher serum sodium levels remained independently associated with in-hospital mortality even when adjusted for gender, years, Stanford type, hypertension, diabetes, coronary artery disease, smoking, drinking, serum glucose, and α-HBDH. Serum sodium ≥139 mmol/L will increase the in-hospital mortality risk in AAD patients.

Serum sodium within or above the normal range, as a common consequence of dehydration and high salt consumption [9], has been associated with decreased left ventricular contractility, increased peripheral insulin resistance, and neuromuscular disturbances [18]. High-sodium intake or an excess of aldosterone increases both local and systemic inflammatory reactions, which are T cell- and macrophage-related [19]. It has been suggested that serum sodium may change vascular function. Oberleithner et al. reported that, in cultured human endothelial cells, cell stiffness increased by 20% within minutes of raising the medium sodium concentration from 135 to 145 mmol/L, which is associated with a reduced nitric oxide formation and endothelial nitric oxide synthase activity, suggesting that an altered plasma sodium concentration may affect vascular endothelial function and thus control vascular tone [20]. Dmitrieva NI et al. [21] also showed that the serum sodium can significantly predict a 10-year

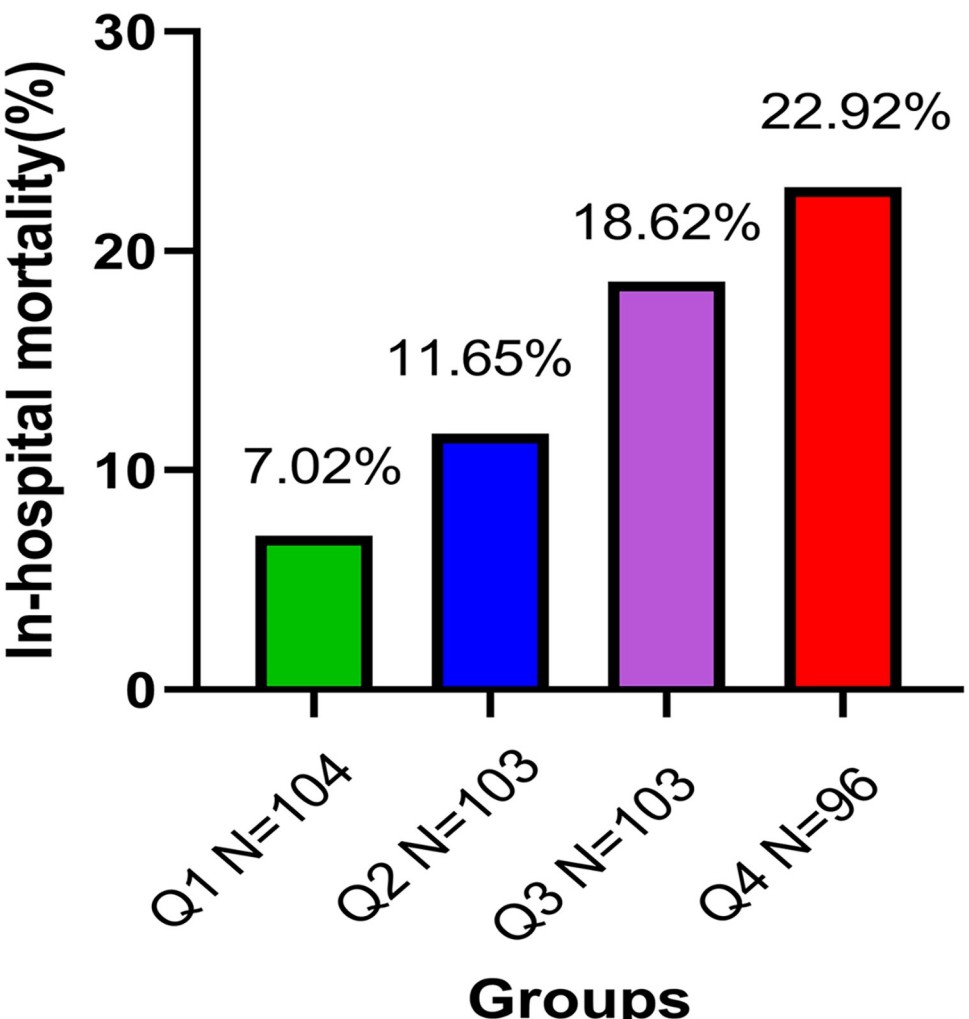

**Fig 3. In-hospital mortality stratified by admission serum sodium quartiles.**

risk of coronary heart disease and demonstrated that slight elevations of extracellular sodium increase the expression of pro-inflammatory mediators in endothelial cells and their adhesive properties, which is accompanied by vascular biology changes, contributing to the development of cardiovascular diseases. Importantly, Anzai et al. [22] found that the massive neutrophil accumulation in the tunica adventitia of the dissected aorta leads to the onset of aortic

**Table 3. Relationship between serum sodium and in-hospital mortality in different models of serum sodium quartiles.**

|  |  | Q1(n = 114) | Q2(n = 103) | Q3(n = 102) | Q4(n = 96) |
|---|---|---|---|---|---|
| in-hospital mortality |  | 8(7.0%) | 12(11.7%) | 19(18.6%) | 22(22.9%) |
| in-hospital mortality(HR) | Crude model | ref | 1.707(0.698–4.178), $P = 0.241$ | 2.890(1.264–6.604), $P = 0.012$ | 3.253(1.447–7.312), $P = 0.004$ |
|  | Model 1 | ref | 1.728(0.705–4.233), $P = 0.232$ | 2.825(1.228–6.502), $P = 0.015$ | 3.241(1.441–7.290), $P = 0.004$ |
|  | Model 2 | ref | 1.718(0.658–4.486), $P = 0.269$ | 3.086(1.242–7.671), $P = 0.015$ | 3.370(1.384–8.204), $P = 0.007$ |

Model 1: Adjusted for age, gender; Model 2: Adjusted for age, gender, Stanford type, hypertension, diabetes, CAD, smoking, drinking, serum glucose, α-HBDH.

For numerical raw data, please see S1 Data.

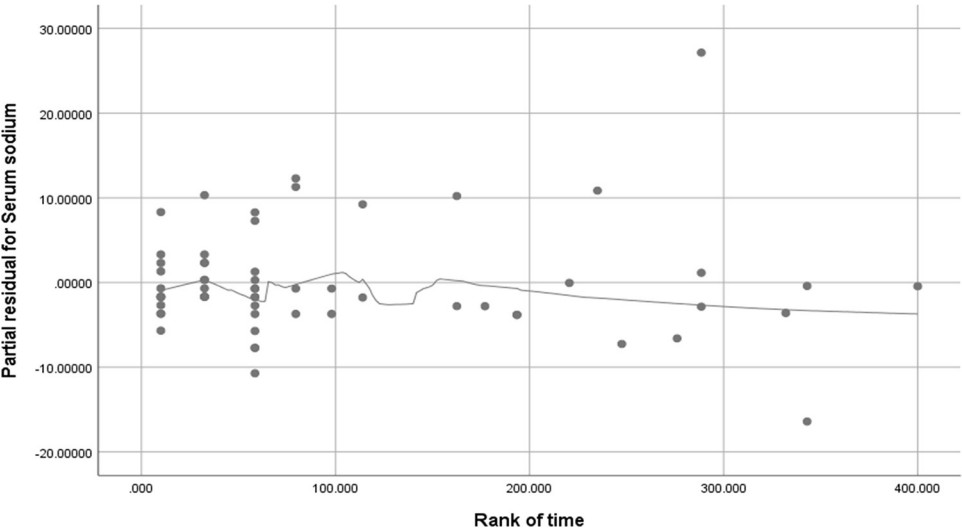

**Fig 4. The smooth curve residual diagram of Schoenfeld's residual with time.**

dissection as well as subsequent aortic rupture [11]. Consistently, in our study, the neutrophil count in AAD patients with the highest serum sodium level (Q4 group) was significantly greater than that in the other three groups, suggesting that serum sodium may provoke and/or promote the formation and development of AAD through neutrophils, but the specific molecular mechanism needs to be further investigated.

Using the quartile for grouped serum sodium, the postoperative in-hospital mortality of AAD patients was higher in the Q3 (139–140 mmol/L) and Q4 ($\geq$ 140 mmol/L) groups than that in the Q1 ($\leq$ 136 mmol/L) group ($P < 0.05$). Tsipotis et al. [7] found that $> 142$ mmol/L serum sodium upon admission increased mortality on unselected in-hospital cohort. Similarly, we also found that the elevated serum sodium ($\geq$ 139 mmol/L) can increase the postoperative in-hospital mortality in AAD patients. Thus, for AAD in-patients, it is not appropriate to take corresponding treatment measures only dependent on the standard of hypernatremia; accurately, managing patients with high serum sodium (139–145 mmol/L) is necessary to improve the prognosis of patients.

Acute type A aortic dissection (ATAAD) is a surgical emergency, with a 90% mortality rate in patients who do not receive timely operative intervention. Despite significant advances in imaging, perioperative care, and surgical technique, ATAAD operative mortality rates have remained relatively unchanged between 10% and 30% over the past two decades [23]. Abdelhamed et al. [24] reported an ATAAD early mortality after surgery of 15.4%. Our results also showed that ATAAD patients had higher in-hospital mortality (22.09%, 57/258), and ATAAD patients with increased serum sodium also had a higher early mortality after surgery (Q1, 9.52%; Q2, 16.42%; Q3, 30.51%; Q4, 31.88%), suggesting that serum sodium may be a surgical risk predictor for early mortality after ATAAD.

There were several limitations of the small cohort of patients in our study. First, this study was a single-center study; a multi-center study is needed in the future to demonstrate serum sodium levels independently associated with in-hospital mortality of AAD patients. Second, the long-term studies for the effect of serum sodium on in-hospital mortality in AAD patients are necessary to better assess and prevent the formation and progress of AAD. Additionally, due to the deaths occurred quickly before or after admission, AAD patients' serum sodium value and other relevant data could not be collected immediately, where 6 hours is the shortest

**Table 4. Univariate and multiple Cox analysis for in-hospital mortality of serum sodium quartiles groups.**

| | Univariate | | | Multivariate | | |
|---|---|---|---|---|---|---|
| | HR | 95%CI | *P* | HR | 95%CI | *P* |
| years | 1.014 | 0.994–1.034 | 0.171 | | | |
| male | 0.678 | 0.414–1.111 | 0.123 | | | |
| Hypertension | 1.312 | 0.684–2.152 | 0.509 | | | |
| Diabetes | 1.004 | 0.246–4.101 | 0.995 | | | |
| CAD | 1.430 | 0.617–3.317 | 0.404 | | | |
| Smoking | 0.768 | 0.474–1.243 | 0.282 | | | |
| Drinking | 0.873 | 9,543–1.405 | 0.576 | | | |
| Stanford type A | 4.503 | 2.048–9.899 | <0.001 | 3.634 | 1.638–8.060 | 0.002 |
| History of trauma | 1.485 | 0.540–4.081 | 0.443 | | | |
| Surgical history | 1.127 | 0.663–1.914 | 0.659 | | | |
| Heart rate | 0.998 | 0.984–1.012 | 0.770 | | | |
| SBP | 0.987 | 0.978–0.995 | 0.002 | | | |
| DBP | 0.986 | 0.973–0.999 | 0.032 | | | |
| ALT | 1.000 | 0.999–1.001 | 0.814 | | | |
| AST | 1.000 | 1.000–1.000 | 0.996 | | | |
| Creatinine | 1.002 | 1.000–1.003 | 0.044 | | | |
| Urea nitrogen | 1.028 | 0.994–1.063 | 0.106 | | | |
| Serum glucose | 1.067 | 1.018–1.117 | 0.007 | 1.077 | 1.025–1.132 | 0.003 |
| α-HBDH | 1.001 | 1.001–1.002 | <0.001 | 1.001 | 1.001–1.002 | <0.001 |
| WBC count | 1.029 | 0.973–1.088 | 0.312 | | | |
| Neutrophil count | 1.032 | 0.975–1.093 | 0.274 | | | |
| lymphocyte count | 0.754 | 0.462–1.228 | 0.256 | | | |
| Monocyte count | 1.405 | 0.738–2.675 | 0.300 | | | |
| Serum sodium | 1.074 | 1.035–1.116 | <0.001 | 1.068 | 1.029–1.109 | 0.001 |
| Serum potassium | 0.999 | 0.983–1.015 | 0.889 | | | |
| Serum chloride | 1.011 | 0.989–1.033 | 0.344 | | | |
| Serum calcium | 0.255 | 0.073–0.899 | 0.034 | | | |

CAD: coronary artery disease; SBP: systolic blood pressure; DBP, diastolic blood pressure; ALT: Alanine transaminase; AST: Aspartate transaminase; α-HBDH: alpha-hydroxybutyric dehydrogenase; WBC: white blood cell.

For numerical raw data, please see S1 Data.

operation interval after admission in our hospital, therefore, some patients who died before surgery were not involved in this study, and these defect data may affect the results of the study.

## Conclusions

Our data indicated that the admission serum sodium would be a potential predictive risk factor for postoperative hospital death of AAD patients. More attention must be paid to high admission serum sodium level in clinic.

## Supporting information

**S1 Data. All numerical raw data are combined in this single excel file.** This file consists of several spreadsheets. Each spreadsheet contains the raw data of one table or one subfigure. (XLSX)

## Acknowledgments

We thank Yong-bo Zhao and Xiao Liu doctors from the Cardiac Surgery Department, the Forth Hospital of Hebei Medical University supporting our data collection work.

## Author Contributions

**Conceptualization:** Dong Ma.

**Data curation:** Pengfei Huang, Dong Ma, Yongbo Zhao, Xiao Liu, Peng Su, Jinjin Zhang, Shuo Ma, Zhe Pan, Juexin Shi, Fangfang Hou, Nana Zhang, Xiaohui Zheng, Nan Liu, Ling Zhang.

**Formal analysis:** Pengfei Huang, Hongyan Wang, Dong Ma.

**Funding acquisition:** Dong Ma.

**Investigation:** Hongyan Wang, Dong Ma, Yongbo Zhao.

**Methodology:** Hongyan Wang, Dong Ma.

**Resources:** Dong Ma.

**Supervision:** Pengfei Huang.

**Validation:** Pengfei Huang.

**Visualization:** Pengfei Huang.

**Writing – original draft:** Hongyan Wang, Dong Ma.

**Writing – review & editing:** Hongyan Wang, Dong Ma.

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
