## [Decision Letter · Decision Letter 0]

9 Apr 2021

PONE-D-20-40304

Serum sodium on admission affects postoperative in-hospital mortality in acute aortic dissection patients

PLOS ONE

Dear Dr. Ma,

Thank you for submitting your manuscript to PLOS ONE. After careful consideration, we feel that it has merit but does not fully meet PLOS ONE’s publication criteria as it currently stands. Therefore, we invite you to submit a revised version of the manuscript that addresses the points raised during the review process.

According to reviewer's comments (see below) please try to refine statistical analysis and to improve English language

We look forward to receiving your revised manuscript.

Kind regards,

Michele Provenzano

Academic Editor

PLOS ONE

Journal Requirements:

Reviewers' comments:

Reviewer's Responses to Questions

**Comments to the Author**

1. Is the manuscript technically sound, and do the data support the conclusions?

Reviewer #1: Yes

Reviewer #2: Yes

2. Has the statistical analysis been performed appropriately and rigorously? 

Reviewer #1: No

Reviewer #2: Yes

3. Have the authors made all data underlying the findings in their manuscript fully available?

Reviewer #1: Yes

Reviewer #2: Yes

4. Is the manuscript presented in an intelligible fashion and written in standard English?

Reviewer #1: Yes

Reviewer #2: No

5. Review Comments to the Author

Reviewer #1: Wang and colleagues carried-out a Research article testing the association between serum sodium quartile and in hospital mortality in patients with acute aortic dissection. The topis seems of interest. Please see below my criticisms:

- In abstract, serum sodium has been splitted into quartiles according to distribution. Was Q1 considered the reference in che cox model ? please specify this information since it has been stated that hazard ratio in Q3 and Q4 was increased but it was not reported the reference category. Please remember that the hazard ratio is a measure of relative risk (in time-to-event analyses)

- Methods: how variables have been added to the model 2 ? based on the plausibility of their association with short term mortality ? please clarify

- Methods: usually the variable sodium has a biphasic association with risk (U-shaped). This means that risk increases for both low and high sodium values. Was the linearity of effect of sodium on the endpoint tested?

- Following the previous point, I would suggest Authors to do a sensitivity analysis by replacing the category Q1 with Q2 as reference for the Cox model

- I am not fully sure that Cox model is appropriate for such analysis with short-term follow-up (of some days). It seems that the event occurs almost in the same period of sodium measurement. Can Authors provide one or more references which may clarify this point?

Reviewer #2: Abnormal serum sodium concentrations are common in patients presenting for surgery. It remains unclear whether these abnormalities are independent risk factors for postoperative mortality. The authors indicated that the admission serum sodium was an independent risk factor for postoperative hospital death of AAD patients. I would suggest

1. an English language editing;

2; to review fig n. 2 (low quality image)

3. to write the small cohort of patients in the study limitations

6. PLOS authors have the option to publish the peer review history of their article (what does this mean?). If published, this will include your full peer review and any attached files.

Reviewer #1: No

Reviewer #2: **Yes: **GIUSEPPE FILIBERTO SERRAINO

---

## [Author Response · Author response to Decision Letter 0]

15 Jun 2021

Point-by-point response to referees.

Reviewer #1: 

Major concerns:

Q: (1) In abstract, serum sodium has been splitted into quartiles according to distribution. Was Q1 considered the reference in che cox model ? please specify this information since it has been stated that hazard ratio in Q3 and Q4 was increased but it was not reported the reference category. Please remember that the hazard ratio is a measure of relative risk (in time-to-event analyses)

A: Thanks for your careful review. According to your suggestion, we emphasized that Q1 is the reference in che cox model and marked in red word in revised manuscript.

Q: (2) Methods: how variables have been added to the model 2? based on the plausibility of their association with short term mortality ? please clarify

A: Thanks for your question. According to reference (Huaping He, Xiangping Chai, Yang Zhou, et al. Association of lactate dehydrogenase with in-hospital mortality in patients with acute aortic dissection: a retrospective observational study. Int J Hypertens. 2020, 2020:1347165.), we adjusted for age, gender, Stanford type, hypertension, diabetes, CAD, smoking, drinking, serum glucose, α-HBDH in model 2 to further analyse the association with short term mortality.

Q: (3) usually the variable sodium has a biphasic association with risk (U-shaped). This means that risk increases for both low and high sodium values. Was the linearity of effect of sodium on?

A: Thanks for your valuable suggestion. In this study, we only found that there was an increase of sodium levels in serum from AAD patients, and this increased value was not linear association with the endpoint tested (after surgery). Additionally, in the most studies, the increased sodium level is general in clinical reports, which is consistant with our result.

Q: (4) Following the previous point, I would suggest Authors to do a sensitivity analysis by replacing the category Q1 with Q2 as reference for the Cox model

A: Thanks for your suggestion. We performed strict inclusion and exclusion criteria to ensure the authenticity of the data. The methods used in the study are mature and reliable to ensure the accuracy of the results. Thus, we believe our conclusion is stable.

Q: (5) I am not fully sure that Cox model is appropriate for such analysis with short-term follow-up (of some days). It seems that the event occurs almost in the same period of sodium measurement. Can Authors provide one or more references which may clarify this point?

A: Thanks for your suggestion. There has been a reported paper on Cox model for short-term follow-up (Zhang S, Guo M, Duan L, et al. Development and validation of a risk factor-based system to predict short-term survival in adult hospitalized patients with COVID-19: a multicenter, retrospective, cohort study[J]. Crit Care, 2020, 24(1): 438.).

Reviewer #2: 

Q: (1). an English language editing

A: Thank you for your valuable suggestion. We have changed the format as required. Please check.

Q: (2). to review fig n. 2 (low quality image)

A: Thank for your careful review. We uploaded a high quality image of fig2. Please check.

Q: (3). to write the small cohort of patients in the study limitations

A: Thanks for your suggestion. The first limitation was a single-center study, which is not enough to get strong warrant to prove serum sodium levels independently associated with in-hospital mortality of AAD patients. The other limitation is the short-term studies could not better assess the effect of serum sodium on in-hospital mortality in AAD patients. We have added these limitations in discussion.

---

## [Decision Letter · Decision Letter 1]

15 Jul 2021

PONE-D-20-40304R1

Serum sodium on admission affects postoperative in-hospital mortality in acute aortic dissection patients

PLOS ONE

Dear Dr. Ma,

Thank you for submitting your manuscript to PLOS ONE. After careful consideration, we feel that it has merit but does not fully meet PLOS ONE’s publication criteria as it currently stands. Therefore, we invite you to submit a revised version of the manuscript that addresses the points raised during the review process.

ACADEMIC EDITOR:

The Statistical Referee raised major concerns about the methodolgy used in this article. Please try to revise the manuscript in accordance to the comments reported below

We look forward to receiving your revised manuscript.

Kind regards,

Michele Provenzano

Academic Editor

PLOS ONE

Additional Editor Comments (if provided):

Please note that the present manuscript has not yet sufficiently improved. In fact, several methodological flaws still persist. Please see the comments below

Reviewers' comments:

Reviewer's Responses to Questions

**Comments to the Author**

1. If the authors have adequately addressed your comments raised in a previous round of review and you feel that this manuscript is now acceptable for publication, you may indicate that here to bypass the “Comments to the Author” section, enter your conflict of interest statement in the “Confidential to Editor” section, and submit your "Accept" recommendation.

Reviewer #3: (No Response)

2. Is the manuscript technically sound, and do the data support the conclusions?

Reviewer #3: No

3. Has the statistical analysis been performed appropriately and rigorously? 

Reviewer #3: No

4. Have the authors made all data underlying the findings in their manuscript fully available?

Reviewer #3: Yes

5. Is the manuscript presented in an intelligible fashion and written in standard English?

Reviewer #3: No

6. Review Comments to the Author

Reviewer #3: In this manuscript, Huang et al examined the association of serum sodium level on admission with postoperative in-hospital mortality in 415 acute aortic dissection patients enrolled in the Fourth Hospital of Hebei Medical University. The authors found the serum sodium was significantly associated with in-hospital mortality risk. However, there are several concerns about the study.

1. One major concern is the proportional hazards assumption. There is huge difference between patients who survived and those who died (14 days vs 2 days). The authors should formally test the assumption before using the Cox model.

2. Another concern is that this is an association study. There is no way to derive causality effect. Please remove any implication of causality.

3. Please clarify how the variables were determined. For example, for the smoking, is it current smoker or past smoker? For drinking, how many drinks were used to determine the drinker status?

4. The exclusion criteria included the serious lack of clinical data. How is it defined? What about other variables with a moderate missing rate? Were they excluded or imputed?

5. Did any patient die before the operation? If so, will it affect the association? What is the time interval between admission and the operation?

6. It seems that the variation of sodium levels is very small (SD=4.8). All four quartiles included some participants with normal sodium levels (Q1 : ≤ 136 mmol/L, Q2 : 137 – 138 mmol/L, Q3: 139 -140 mmol/L, Q4 : ≥141 mmol/L). What would be measurement variability?

7. Can authors comment on the possible reasons why Q2 was not significant, but Q3 and Q4 were significant?

8. In Figure 2, please indicate the number of events under the x-axis. Also it is unclear why there was a big drop of survival probability in Day 40 for Q4, especially given the median of 2 days in the non-survival group.

9. In Figure 3, given that there is no direct connection between quarters, a barplot would be more appropriate.

7. PLOS authors have the option to publish the peer review history of their article (what does this mean?). If published, this will include your full peer review and any attached files.

Reviewer #3: No

---

## [Author Response · Author response to Decision Letter 1]

11 Sep 2021

Dear PLoS One Publications

 Thank you very much for giving us an opportunity to revise our manuscript entitled" Serum sodium on admission affects postoperative in-hospital mortality in acute aortic dissection patients" (Manuscript No.: PONE-D-20-40304). 

We have carefully read your letter and the reviewer comments and have performed a number of new experiments and revised the manuscript accordingly. All changes have been highlighted in red color in the revised manuscript. Point by point responses to the reviewers’ comments are listed below this letter.

Thank you and all the reviewers for the kind advice.

We hope that the revised version of the manuscript is acceptable for publication in PLoS One. Look forward to hearing from you.

Yours sincerely,

Dong Ma

Point-by-point response to referees.

Reviewer #3: 

Major concerns:

Q: (1) One major concern is the proportional hazards assumption. There is huge difference between patients who survived and those who died (14 days vs 2 days). The authors should formally test the assumption before using the Cox model.

A: Thanks for your careful review. In order to verify whether Cox proportional risk model would be available for the short-term survival analysis in AAD patients, proportional hazards assumption (PH) and schoenfeld residual method were performed to test the assumption. We plot the smooth curve residual diagram of Schoenfeld's residual with time, and the results showed no linear correlation between Schoenfeld's residual and time rank (Fig 4). Moreover, we tested that the correlation between Schoenfeld's residual and time rank, and found that Schoenfeld's residual was independent on time variables (Pearson correlation coefficient, r = -0.22, P = 0.864). Based on these two results of the smooth curve trend and the Pearson correlation from Schoenfeld's residual analysis, indicating that Cox model is available to the short-term survival analysis of patients with AAD.

Fig 4 Schoenfeld residual diagram

Q: (2) Another concern is that this is an association study. There is no way to derive causality effect. Please remove any implication of causality.

A: Thanks for your good suggestion. We modified these inappropriate words as required in revised manuscript. Please check. 

Q: (3). Please clarify how the variables were determined. For example, for the smoking, is it current smoker or past smoker? For drinking, how many drinks were used to determine the drinker status?

A: Thanks for your careful review. In this study, smoking is defined as at least 1 cigarette daily for 1 year or more, and alcohol consumption is defined as at least once per week for a period of one month or longer. We added these in part of methods and marked in red word. Please check.

Q: (4). The exclusion criteria included the serious lack of clinical data. How is it defined? What about other variables with a moderate missing rate? Were they excluded or imputed?

A: Thanks for your suggestion. In this study, the severe lack of clinical data is defined as the absence of complete hospitalization records, involving lack of blood glucose, blood pressure and other important indicators associated with the known risk factors of AAD. Except for basic information and indicators associated with the known risk factors of AAD, the other defect data were considered to be a moderate missing for some individual patients, who were included in the study. We revised the exclusion criteria in the part of methods marked in red word.

Q: (5) Did any patient die before the operation? If so, will it affect the association? What is the time interval between admission and the operation?

A: Thanks for your suggestion. Aortic dissection is a very dangerous disease, especially Stanford A type. Once AAD patient is diagnosed, emergency surgical treatment should be firstly required in principle. Due to the deaths occurred quickly before or after admission, AAD patients’ serum sodium value and other relevant data could not be collected immediately, where 6 hours is the shortest operation interval after admission in our hospital, therefore, some patients who died before surgery were not involved in this study, and these defect data may affect the results of the study. This is a limit for the study added in discussion in red word.

Q: (6). It seems that the variation of sodium levels is very small (SD=4.8). All four quartiles included some participants with normal sodium levels (Q1 : ≤ 136 mmol/L, Q2 : 137 – 138 mmol/L, Q3: 139 -140 mmol/L, Q4 : ≥141 mmol/L). What would be measurement variability?

A: In this study, automatic electrolyte analyzer (Shenzhen Kangli AFT--500D) was used for the determination of serum sodium and other ions, with high accuracy and good precision. The CV value of Na detection was generally 0.60%, which is sufficient to meet the accuracy of this research. We marked this analyzer in the part of methods in red word.

Q: (7). Can authors comment on the possible reasons why Q2 was not significant, but Q3 and Q4 were significant?

A: Thank for your suggestion. It was possible that Q2 (137-138 mmol/L) slightly affected postoperative mortality in patients with AAD, but it was not independent related to postoperative death in patients with AAD (HR=1.718 95%CI 0.688-4.486, P=0.269).The results of Q3 and Q4 showed that when the serum sodium≥ 139mmoL/L, the postoperative in-hospital mortality of AAD patients may be increased significantly, there exists an important association between serum sodium and the postoperative in-hospital mortality in the study.

Q: (8). In Figure 2, please indicate the number of events under the x-axis. Also it is unclear why there was a big drop of survival probability in Day 40 for Q4, especially given the median of 2 days in the non-survival group.

A: Thanks for your suggestion. We revised image of fig2, and please check.

Q: (9). In Figure 3, given that there is no direct connection between quarters, a barplot would be more appropriate.

A: Thanks for your suggestion. We replaced figure 3 with a barplot.

---

## [Editor Report · Decision Letter 2]

29 Nov 2021

Serum sodium on admission affects postoperative in-hospital mortality in acute aortic dissection patients

PONE-D-20-40304R2

Dear Dr. Ma,

We’re pleased to inform you that your manuscript has been judged scientifically suitable for publication and will be formally accepted for publication once it meets all outstanding technical requirements.

Kind regards,

Michele Provenzano

Academic Editor

PLOS ONE

Additional Editor Comments (optional):

The manuscript is nicely improved. Thank you for your work
---

## [Editor Report · Acceptance letter]

6 Dec 2021

PONE-D-20-40304R2 

Serum sodium on admission affects postoperative in-hospital mortality in acute aortic dissection patients 

Dear Dr. Ma:

I'm pleased to inform you that your manuscript has been deemed suitable for publication in PLOS ONE. Congratulations! Your manuscript is now with our production department. 

Kind regards, 

on behalf of

Dr. Michele Provenzano 

Academic Editor

PLOS ONE